# Deep inference of latent dynamics with spatio-temporal super-resolution using selective backpropagation through time

**Feng Zhu*[1], Andrew R. Sedler*[2], Harrison A. Grier[3], Nauman Ahad[4], Mark A. Davenport[4], Matthew T. Kaufman[5,6], Andrea Giovannucci[7,8,9], and Chethan Pandarinath[10,11]**

[1]Neuroscience Graduate Program, Emory University
[2]Center for Machine Learning, Georgia Tech
[3]Computational Neuroscience Graduate Program, The University of Chicago
[4]School of Electrical and Computer Engineering, Georgia Tech
[5]Dept. of Organismal Biology and Anatomy, The University of Chicago
[6]Neuroscience Institute, The University of Chicago
[7]Joint Dept. of Biomedical Engineering, UNC Chapel Hill and NC State University
[8]UNC Chapel Hill Neuroscience Center
[9]NC State/UNC Closed-Loop Engineering for Advanced Rehabilitation (CLEAR)
[10]Coulter Dept. of Biomedical Engineering, Emory University and Georgia Tech
[11]Dept. of Neurosurgery, Emory University

## Abstract

Modern neural interfaces allow access to the activity of up to a million neurons within brain circuits. However, bandwidth limits often create a trade-off between greater spatial sampling (more channels or pixels) and the temporal frequency of sampling. Here we demonstrate that it is possible to obtain spatio-temporal super-resolution in neuronal time series by exploiting relationships among neurons, embedded in latent low-dimensional population dynamics. Our novel neural network training strategy, selective backpropagation through time (SBTT), enables learning of deep generative models of latent dynamics from data in which the set of observed variables changes at each time step. The resulting models are able to infer activity for missing samples by combining observations with learned latent dynamics. We test SBTT applied to sequential autoencoders and demonstrate more efficient and higher-fidelity characterization of neural population dynamics in electrophysiological and calcium imaging data. In electrophysiology, SBTT enables accurate inference of neuronal population dynamics with lower interface bandwidths, providing an avenue to significant power savings for implanted neuroelectronic interfaces. In applications to two-photon calcium imaging, SBTT accurately uncovers high-frequency temporal structure underlying neural population activity, substantially outperforming the current state-of-the-art. Finally, we demonstrate that performance could be further improved by using limited, high-bandwidth sampling to pretrain dynamics models, and then using SBTT to adapt these models for sparsely-sampled data.

35th Conference on Neural Information Processing Systems (NeurIPS 2021).

* Contributed equally. Corresponding authors: `fzhu23@emory.edu`, `{asedler,chethan}@gatech.edu`

# 1  Introduction

Modern systems neuroscientists have access to the activity of many thousands to potentially millions of neurons via multi-photon calcium imaging and high-density silicon probes [1–4]. Such interfaces provide a qualitatively different picture of brain activity than was achievable even a decade ago.

However, neural interfaces increasingly face a trade-off – the number of neurons that can be accessed (capacity) is often far greater than the number that is simultaneously monitored (bandwidth). For example, with 2-photon calcium imaging (2p; **Fig. 1a**, *top*), hundreds to thousands of neurons are serially scanned by a laser that traverses the field of view, resulting in different neurons being sampled at different times within an imaging frame. As a consequence, a trade-off exists between the size of the field-of-view (and hence the number of neurons monitored), the sampling frequency, and the signal-to-noise with which each neuron is sampled. Whereas current analysis methods treat 2p data as if all neurons within a field-of-view were sampled at the same time at the imaging frame rate, the fact that each neuron is sampled at staggered, known times within the frame could be employed to increase the time resolution.

Electrophysiological interfaces face similar trade-offs (**Fig. 1a**, *bottom*). With groundbreaking high-density probes such as Neuropixels and Neuroseeker [3–5], simultaneous monitoring of all recording sites is either not currently possible or limits the signal-to-noise ratio, so users typically monitor a selected subset of sites within a given recording session. For example, Neuropixels 2.0 probes contain up to 5120 electrodes, 384 of which can be recorded simultaneously [4]. In other situations, power constraints might make it preferable to restrict the number of channels that are simultaneously monitored, such as in wireless or fully-implanted applications where battery life and heat dissipation are key challenges [6–8]. As newer interfacing strategies provide a pathway to hundreds of thousands of channels for revolutionary brain-machine interfaces [9, 10], neural data processing strategies that can leverage dynamic deployment of recording bandwidth might allow substantial power savings.

Solutions to these space-time trade-offs may come from the structure of neural activity itself. A large body of work suggests that the activity of individual neurons within a large population is not independent, but instead is coordinated through a lower-dimensional, latent state that evolves with stereotyped temporal structure (**Fig. 1b**). We can represent the state at time $t$ as a vector $\boldsymbol{x}_t \in \mathbb{R}^D$ that evolves according to dynamics captured by a function $f$ such that $\boldsymbol{x}_{t+1} \approx f(\boldsymbol{x}_t)$. Rather than

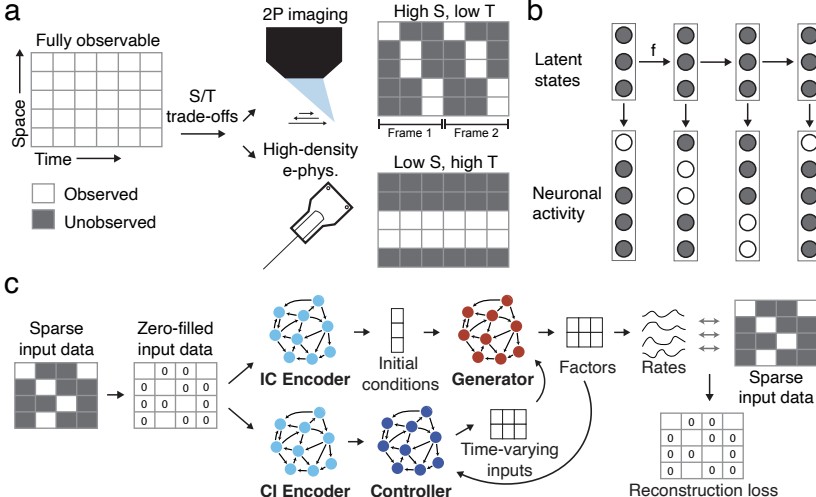

Figure 1: Exploiting space-time trade-offs in neural interfaces using SBTT. (a) In 2-photon calcium imaging (top), individual neurons are serially scanned at a low frame rate, resulting in staggered sample times. In modern electrophysiological recordings (bottom), bandwidth or power constraints prevent simultaneous monitoring of all recording sites. (b) Observed neuronal activity reflects latent, low-dimensional dynamics (captured by the function $f$). (c) SBTT applied to a sequential autoencoder for inferring latent dynamics from neural population activity.

directly observing the latent state $x_t$, we observe neural activity that we represent as $y_t \in \mathbb{R}^N$, where $y_t \approx h(x_t)$ for some function $h$. Due both to the fact that $f$ imposes a significant amount of structure on the trajectory of the $x_t$'s and the fact that we typically expect the dimension $D$ of $x_t$ to be far smaller than the number of possible observations $N$, one might expect that it should be possible to estimate the $x_t$'s without observing every neuron at every time step (i.e., measuring only some of the elements of each $y_t$), just as we generally infer latent states from only a fraction of the neurons in a given area. If so, principled exploitation of the space-time trade-off of neural interfaces might achieve higher-fidelity or more bandwidth-efficient characterization of neural population activity.

To our knowledge, no methods have demonstrated inference of dynamics from data in which the set of neurons being monitored changes dynamically at short intervals. To address this challenge, we introduce *selective backpropagation through time* (SBTT; **Fig. 1c**), a method to train deep generative models of latent dynamics from data where the identity of observed variables varies from sample to sample. Here we explore applications of SBTT to state space modeling of neural population activity that obeys low-dimensional dynamics.

This paper is organized as follows. Section 2 provides an overview of related work. Section 3 details SBTT and its integration with sequential autoencoders for modeling neural population dynamics. Section 4 demonstrates the effectiveness of this solution in achieving more efficient and higher-fidelity inference of latent dynamics in applications to electrophysiological and calcium imaging data.

## 2  Related work

There is a long and rich literature on methods for system identification, particularly in the case of *linear* dynamical systems. The last several years have witnessed a burst of activity in establishing a more robust theoretical understanding of when and how well these methods work. Particularly relevant to our approach, [11] shows that under suitable conditions on the dynamical system, performing gradient descent on the reconstruction loss of observed data can provably recover the parameters of the system despite the nonconvexity of the problem. Additional guarantees are provided in [12–16] which make varying assumptions on the underlying dynamics and the observation function, the existence of an observable control input, and the stochasticity of the dynamical system. Adversarial noise models are further considered in [17, 18]. We emphasize, however, that all of the above works limit their focus to *linear* dynamical systems where the observations are *fully sampled*, i.e., where all of $y_t = Hx_t$ is measured for all $t$.

In the case of a linear observation model ($y_t = Hx_t$) but where we observe only a subset of the elements of each $y_t$, the problem is reminiscent of the *low-rank matrix completion* problem [19]. Specifically, by letting $Y$ and $X$ denote the matrices whose columns are given by the $y_t$ and $x_t$ respectively, we can write $Y = HX$. If $D \ll N$, this is a low-rank matrix, and hence could be recovered from a random sampling of $O(D \log N)$ elements of each column of $Y$ [19]. However, this strategy essentially assumes that there is no relationship between the $x_t$ – one would expect to obtain significant improvements by exploiting the dynamical structure among the $x_t$ imposed by $f$. Indeed, in [20, 21] the authors show that if the dynamics $f$ are *known*, then it is possible to significantly reduce the sampling requirements. However, the question of *learning* such an $f$ from undersampled observations has again not been addressed in this literature.

In some application domains, there have been hints in this direction. In particular, in the related contexts of recommendation systems [22, 23] and student knowledge tracking [24, 25] there have been successful empirical efforts aimed at learning dynamical systems for modeling how user preferences/knowledge change over time. While such approaches have also had to confront the issue of missing observations (items that are not rated or questions that are not answered), they are aided by the existence of rich sources of additional metadata (e.g., tags) that lead to fundamentally different approaches than what we take here.

Within our application domain, a variety of methods have been developed to infer latent dynamical structure from neural population activity on individual trials, including those based on Gaussian processes [26–29], linear [30–32] and switching linear dynamical systems [33–35], and nonlinear dynamical systems such as recurrent neural networks [36–39], hidden Markov models [40], neural ODEs [41], and transformers [42]. Variants of these methods accommodate cases where the particular observed neurons change over long time periods (e.g., over the course of days) [36, 43, 44], but these are not appropriate for cases where neurons are intermittently sampled on short timescales. As

described below, several of these methods would be amenable to using SBTT to adapt to intermittent sampling, as SBTT should be applicable to any neural network architecture that learns weights via backpropagation through time.

# 3 Selective backpropagation through time

## 3.1 Overview

SBTT is a learning rule for updating the weights of a neural network that allows backpropagation of loss for the portions of data that are present while preventing missing data from corrupting the gradient signal. The technique optimizes the model to reconstruct observed data while extrapolating to the unobserved data. The implementation of SBTT is related to other approaches that augment network inputs and cost functions to reflect different subsets of the data matrix across samples, in particular coordinated dropout [37], masked language modeling [45], and DeepInterpolation [46]. Though not designed for missing data, these previous approaches split fully-observed data into two portions - a portion that is provided at the input to the network, and a portion that is used to compute loss at the output. SBTT uses a similar strategy to accommodate missing data, by zero-filling missing input points and aggregating only losses for observed data points at the output. Though prior work has adopted a similar strategy to handle missing data [47], the contribution of SBTT is integrating the strategy with models with a temporal component to learn a dynamical system. To demonstrate SBTT, we provide code for a basic experiment using a sequential autoencoder and Lorenz dataset (`https://github.com/snel-repo/sbtt-demo`).

## 3.2 Illustration with a simple linear dynamical system

We begin by describing our approach in the context of a simple linear dynamical system. In the case where we have no (observable) inputs, we can model a linear dynamical system as

$$\boldsymbol{x}_{t+1} = \boldsymbol{A}\boldsymbol{x}_t + \boldsymbol{w}_t$$
$$\boldsymbol{y}_t = \boldsymbol{H}\boldsymbol{x}_t + \boldsymbol{z}_t.$$

Here, $\boldsymbol{x} \in \mathbb{R}^D$ represents a hidden state, $\boldsymbol{y} \in \mathbb{R}^N$ represents our observations, and $\boldsymbol{w}_t$ and $\boldsymbol{z}_t$ represent noise. The matrix $\boldsymbol{A}$ models the dynamics of the hidden state, and $\boldsymbol{H}$ models the observation function of our system. In this setting, our task is to learn the parameters $\boldsymbol{A}$ and $\boldsymbol{H}$ given the observations $\boldsymbol{y}_0, \ldots, \boldsymbol{y}_{T-1}$ as well as the initial system state $\boldsymbol{x}_0$.

SBTT is a variation of standard back-propagation where loss terms attributed to missing observations are ignored when computing back-propagation updates. Concretely speaking, consider a linear recurrent network that can learn this linear model using a least squares loss

$$\mathcal{L} = \frac{1}{T} \sum_{t=0}^{T-1} \frac{1}{2} \|\boldsymbol{y}_t - \boldsymbol{H}\boldsymbol{x}_t\|_2^2.$$

If the observation vector $\boldsymbol{y}_t$ contains a missing entry at index $i$, the least squares loss would not contain the $(y_t^i - (Hx_t)^i)^2$ term, where the superscript $i$ represents the $i$th index of a vector. If $\boldsymbol{o}_t = \boldsymbol{H}\boldsymbol{x}_t$ is taken to be the output of the recurrent network at time step $t$, then the loss with respect to the outputs of the network is

$$\frac{\partial \mathcal{L}}{\partial \boldsymbol{o}_t} = \frac{1}{T}(\boldsymbol{o}_t - \boldsymbol{y}_t). \tag{1}$$

SBTT requires that loss terms, and subsequently loss gradients, related to missing observations are ignored. This means that elements in the gradient vector (1) are ignored and set to 0 at indices $i$ where the corresponding observations, $y_t^i$, are missing. This gradient is then back-propagated through time to obtain gradients with respect to model parameters $\boldsymbol{A}$ and $\boldsymbol{H}$ as shown below

$$\frac{\partial \mathcal{L}}{\partial \boldsymbol{H}} = \sum_{t=0}^{T-1} \frac{\partial \mathcal{L}}{\partial \boldsymbol{o}_t}(\boldsymbol{x}_t)^\mathsf{T}, \frac{\partial \mathcal{L}}{\partial \boldsymbol{A}} = \sum_{t=1}^{T-1} \frac{\partial \mathcal{L}}{\partial \boldsymbol{x}_t} \boldsymbol{x}_{t-1}^\mathsf{T},$$

where $\frac{\partial L}{\partial \boldsymbol{x}_t}$ is recursively computed using back-propagation through time:

$$\frac{\partial \mathcal{L}}{\partial \boldsymbol{x}_t} = \boldsymbol{A}^\mathsf{T} \frac{\partial \mathcal{L}}{\partial \boldsymbol{x}_{t+1}} + \boldsymbol{H}^\mathsf{T} \frac{\partial \mathcal{L}}{\partial \boldsymbol{o}_t}.$$

These parameters can then be updated using gradient descent.

### 3.3 Integration with a deep generative model of neural population dynamics

Here we will demonstrate the use of SBTT with a recently developed framework for inferring nonlinear latent dynamics from neural population recordings. This framework, Latent Factor Analysis via Dynamical Systems (LFADS), is a sequential variational auto-encoder (SVAE), detailed in [36]. LFADS models single-trial latent dynamics by learning the initial state of the dynamical system, the dynamical rules that govern state evolution, and any time-varying inputs that cannot be explained by the dynamics (i.e., in the case of a non-autonomous dynamical system). Briefly, a bidirectional RNN encoder operates on the neural spiking sequence $\mathbf{y}(t)$ and produces a conditional distribution over initial condition $\mathbf{z}$, $Q(\mathbf{z}|\mathbf{y}(t))$. A Kullback-Leibler (KL) divergence penalty is applied as a regularizer for divergence between the uninformative prior $P(\mathbf{z})$ and $Q(\mathbf{z}|\mathbf{y}(t))$. The initial condition is then drawn from $Q(\mathbf{z}|\mathbf{y}(t))$ and mapped to an initial state for a generator RNN, which learns to approximate the dynamical rules underlying the neural data. A controller RNN takes as input the state of the generator at each time step, along with a time-varying encoding of $\mathbf{y}(t)$ (produced by a second bidirectional RNN encoder), and injects a time-varying input $\mathbf{u}(t)$ into the generator. Similar to $\mathbf{z}$, $\mathbf{u}(t)$ is drawn from a parameterized time-varying distribution of $Q(\mathbf{u}(t)|\mathbf{y}(t))$ produced by the controller. A second KL penalty is applied between $P(\mathbf{u}(t))$ and $Q(\mathbf{u}(t)|\mathbf{y}(t))$. At each time step, the generator state evolves with input from the controller and the controller receives delayed feedback from the generator. The generator states are linearly mapped to factors, which are in turn mapped to the firing rates of the neurons using a linear mapping followed by an exponential nonlinearity. LFADS assumes a Possion emission model for the observed spiking activity. The optimization objective combines the reconstruction cost of the observed spiking activity (i.e., the Poisson likelihood of the observed spiking activity given the rates produced by the generator network), the KL penalties described above, and L2 regularization penalties on the weights of the recurrent networks. During training, network weights are optimized using stochastic gradient descent and backpropagation through time.

The first step in applying SBTT to LFADS is to zero-fill the missing data before feeding it into the initial condition (IC) and controller input (CI) encoders. After passing the data through the remaining hidden layers, we use the resulting rate estimates to compute a reconstruction loss (Poisson negative log-likelihood) for each observed neuron-timepoint and aggregate by taking the mean. The modified reconstruction loss is combined with other losses as in the standard LFADS model. The network only optimizes for reconstruction of observed data and is free to interpolate at unobserved points.

Throughout this paper we use population-based training along with coordinated dropout, together known as AutoLFADS, to optimize our models [37, 39, 48]. This framework is essential for achieving reliably high-performing LFADS models, regardless of dataset statistics. Hyperparameters, search ranges, and training details are given in the supplement.

## 4 Experiments

### 4.1 High performance with limited bandwidth on primate electrophysiological recordings

A key target application of AutoLFADS with SBTT is to enable reduced sampling of electrodes: either to enable recording from larger populations of electrodes with limited bandwidth (such as with Neuropixels), or to reduce power consumption (such as for fully-implantable brain-machine interfaces). To investigate the performance of AutoLFADS models trained with SBTT, we started with a large and well-characterized dataset containing electrophysiological recordings from macaque primary motor and dorsal premotor cortex (M1/PMd) [49, 50]. The data were collected during a delayed reaching task, in which the monkey made both straight and curved reaches from a center position, around virtual barriers (the maze), to one of 108 possible target positions. The dataset consisted of 2296 trials with 202 sorted units aligned to movement onset in a window from 250 ms before to 450 ms after this point. Spike counts were binned at 10 ms (70 bins). We held out 50 randomly selected units from modeling to use for evaluation of inferred latent factors. We simulated various missing data scenarios for the remaining 152 units by randomly masking a fraction of the observations at each time step for each trial (**Fig. 2a**, *top*).

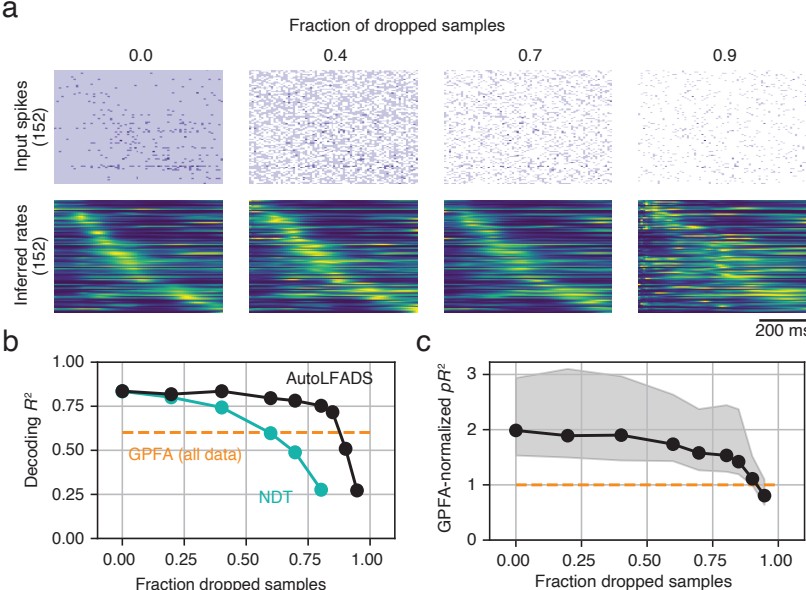

Figure 2: SBTT allows inference of latent dynamics from M1/PMd electrophysiology data with sparse observations. (a) Spike count input and inferred rate output of LFADS for the same example trial with increasingly sparse observations. Masked data are shown in white, observed zeroes are shown in light purple, and nonzero spike counts are shown in darker shades. Units are sorted by timing of firing rate peaks for the fully sampled model. (b) Accuracy of linear hand velocity decoding from inferred latent factors. (c) Quality of GLM fits from inferred latent factors to 50 held-out units. $pR^2$ values for each held-out unit are normalized to the corresponding values achieved by the GPFA baseline. Points denote the median across all units. Shaded areas depict the 25th and 75th quantiles.

For each of the masked datasets, we used AutoLFADS with SBTT to robustly train neural dynamics models. Latent factors and firing rates were inferred for all time steps, despite the missing (masked) observations. Even with 70% dropped samples, the inferred firing rates showed structure comparable to the model of fully observed data (**Fig. 2a**, *bottom*).

To determine whether the models were able to capture biologically relevant information from sparsely sampled data, we evaluated the inferred latent factors in terms of their ability to predict hand velocity (**Fig. 2b**) and the spiking activity of held-out units (**Fig. 2c**). As a recognizable baseline, we trained a Gaussian Process Factor Analysis (GPFA) model (40 latent dimensions, 20 ms bins) on the fully observed dataset [26, 51]. GPFA is a commonly-used and versatile method for extracting latent structure from neural population activity, and these parameters have been validated on this dataset in prior work [36]. We trained simple linear decoders to predict hand velocity from the inferred latent factors with an 80 ms delay (50/50, trial-wise train-test split), and evaluated using the coefficient of determination, averaged over x- and y-dimensions. For AutoLFADS with SBTT, decoding performance showed a minimal decline until around 80% of the data had been dropped, with some models outperforming the GPFA baseline using as little as 15% of the original data (**Fig. 2b**). To measure how well the models captured the population structure, we trained generalized linear models (GLMs) [52, 53] to predict the spikes for the held out units and evaluated fit quality using pseudo-$R^2$ ($pR^2$). Similar to the decoding results, we found that AutoLFADS with SBTT captured population structure significantly better than fully observed GPFA, and that the information content of the factors declined slowly until about 80% missing samples (**Fig. 2c**). More detail on the $R^2$ and $pR^2$ metrics can be found in Supplement Section E.

To evaluate the importance of modeling latent dynamics for accurate inference with sparsely observed data, we also trained NDT with selective backpropagation on the same datasets [42]. We found that decoding performance from inferred firing rates declined faster than for AutoLFADS with SBTT, but NDT still outperformed GPFA with up to 40% missing data (**Fig. 2b**).

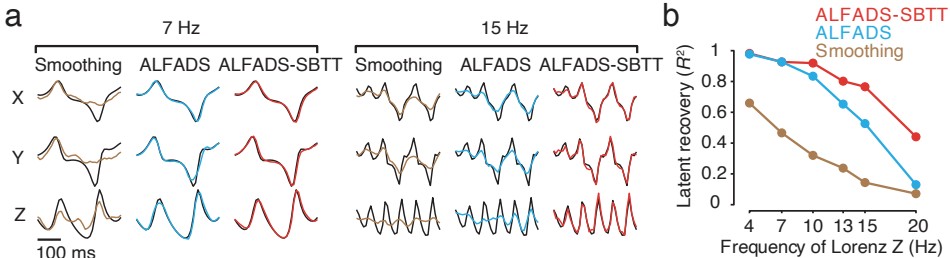

Figure 3: SBTT improves inference of high-frequency dynamics from simulated 2P data with known dynamical structure. (a) True and inferred Lorenz latent states (X/Y/Z dimensions) for a single example trial from Lorenz systems simulated at two different frequencies (7Hz and 15Hz). Black: ground truth. Colored: inferred. (c) Performance in estimating the Lorenz Z dimension as a function of Lorenz speed was quantified by variance explained ($R^2$) for all three methods. The speed of the Lorenz dynamics was quantified based on the peak location of the power spectra of Lorenz Z dimension, with a sampling frequency of 100Hz.

## 4.2 Recovery of high frequency features in simulated 2P calcium imaging data

High-frequency features of neural responses are generally assumed to be lost in 2P imaging due to limited scanning speeds and indicator kinetics. We hypothesized that some of the loss is actually due to standard 2P data processing, which discards information regarding sub-frame sampling time of individual neurons, and that SBTT could recover some of this information. The inherently staggered sampling of neurons due to raster scanning can be treated as a time series with missing values and higher temporal resolution than the frame rate. We tested SBTT on both simulated and real calcium imaging data. In both cases, we adapted AutoLFADS to better account for the statistics of deconvolved calcium activity (AutoLFADS-ZIG, see supplement) by substituting the underlying Poisson emission model with a Zero-Inflated Gamma distribution [54]. In our experiments we compared three methods: AutoLFADS-ZIG with SBTT (ALFADS-SBTT), a standard frame-resolution version of AutoLFADS-ZIG without SBTT (ALFADS), and Gaussian smoothing of deconvolved calcium activity.

We generated artificial 2P data from a population of simulated neurons (278 neurons) whose firing rates were linked to the state of an underlying Lorenz system [28, 55] (see supplement). To assess the ability to reconstruct latent dynamics at different frequencies, we simulated Lorenz systems with different speeds. For each Lorenz system we report the Z dimension power spectrum peak, which contains the most concentrated and highest frequencies. Fluorescence traces were simulated from the spike trains using an order 1 autoregressive model followed by a non-linearity and injected with 4 sources of noise (see supplement). Firing rates were simulated with a sampling frequency of 100Hz, and a "location" was randomly chosen for each simulated neuron, such that sampling times for different neurons were staggered to simulate 2p laser scanning sampling times. This produced fluorescence traces with one of three possible associated phases (0,11,22ms) and overall sample rate 33 Hz. We deconvolved neural activity from the fluorescence traces using the OASIS algorithm [56] as implemented in the CaImAn package [57].

For ALFADS-SBTT we used the sub-frame phase information to generate intermittently-sampled data. In contrast, for both ALFADS and Gaussian smoothing, we discarded phase information and collapsed samples into a single time bin per frame, as is standard in 2p imaging data processing. To evaluate the performance in recovering the ground truth Lorenz states, we trained a mapping from the output of each method (i.e., the inferred event rates from ALFADS-SBTT and ALFADS, and smoothed deconvolved events by Gaussian smoothing; signals were interpolated to 100 Hz for the latter methods) to the ground truth Lorenz states using cross-validated ridge regression. We used $R^2$ between the true and inferred Lorenz states as a metric of performance.

The true and predicted Lorenz states for two example trials are illustrated in **Fig. 3a**. The performance of smoothing and A-FR dropped substantially for higher Lorenz state frequencies, while A-SBTT maintained reasonable estimates ($R^2 \approx 0.8$) up to 15Hz (**Fig. 3a & b**) and never dropped below $0.4$ in the range of tested frequencies.

## 4.3 Improved representation of hand kinematics in mouse 2P calcium imaging data

We next applied SBTT to real 2P calcium imaging data we collected from motor cortex in a mouse performing a forelimb water grab task. The dataset comprised 475 trials in which the mouse was cued by a tone to reach to a left or right spout and retrieve a droplet of water with its right forepaw. Pyramidal cells expressing the GCaMP6f calcium indicator were imaged with a two-photon microscope at a 31 Hz frame rate, and a subset of 439 modulated neurons within the field-of-view (FOV) were considered for analysis (FOV shown in **Fig. 4a**, *left*; example calcium traces in **Fig. 4a**, *right*). The mouse's forepaw position was tracked in 3D at 150 Hz with stereo cameras and DeepLabCut [58]. Calcium events were deconvolved with OASIS [56, 57].

2P data for ALFADS-SBTT were processed analogously to the simulations, using neuron locations within the FOV to inform the intermittent sampling times. Trials represented a window spanning 200 ms before to 800 ms after the mouse's reach onset. This resulted in 100 time points per trial for ALFADS-SBTT, and 31 time points per trial for ALFADS and Gaussian smoothing. For both ALFADS-SBTT and ALFADS, trials were split into 80/20 train/validation.

To compare representations inferred by ALFADS-SBTT and ALFADS, we first evaluated how closely the single-trial event rates inferred for each neuron resembled that neuron's peri-stimulus time histogram (PSTH). PSTHs were calculated by taking the average of the Gaussian-smoothed deconvolved events across trials within each experimental condition. Because the mouse's reaches were not stereotyped to each spout (i.e., left or right), we subgrouped trials into 4 finer conditions based on forepaw Z position during the reach. ALFADS-SBTT single-trial event rates were more strongly correlated with neurons' PSTHs compared to those inferred by ALFADS (**Fig. 4b**).

We next decoded the mouse's single-trial forepaw kinematics (position and velocity) based on each model's output. Decoding was performed using ridge regression with 5-fold cross validation. We used $R^2$ between the true and predicted hand positions and velocities as a metric of performance. $R^2$ was averaged across XYZ behavioral dimensions and all 5 folds of the test sets. Decoding using ALFADS-SBTT inferred rates outperformed results from smoothing deconvolved events, or from the ALFADS inferred rates (**Fig. 4c**). Because the improvement of decoding performance for position is modest, we further assessed how the improvement was distributed as a function of temporal frequency. We computed the coherence between the true and decoded positions for each method (**Fig. 4d**). Consistent with the simulations, ALFADS-SBTT predictions showed higher coherence with true position than predictions from other methods, with improvements more prominent at higher frequencies (5-15Hz).

## 4.4 Using high-bandwidth observations to improve performance in low-bandwidth conditions

In implantable or wireless applications, using the device's full interface bandwidth might incur significant power costs, which would burden users with frequent battery recharging. However, it may be possible to leverage high-bandwidth recordings from limited time periods to learn models of latent dynamics, and then switch to low-bandwidth modes for subsequent long-term operation, in order to minimize ongoing power use. Such an approach is enabled by the stability of latent dynamics over months to years [36, 44, 59].

We tested these ideas on the same electrophysiological dataset described in section 4.1. After training AutoLFADS models on the fully sampled data, we retrained the initial condition and controller input encoders using SBTT on each of the sparsely sampled datasets. The weights for the rest of the network remained fixed. In this way, the dynamical rules learned from the fully sampled data are maintained, while the mappings from data to the initial conditions and controller inputs are adapted for sparse data. Retraining the encoding networks in this way (**Fig. 5**, "Retrained sparse") maintained performance to high levels of missing data, outperforming AutoLFADS trained on fully observed data but run with missing data (**Fig. 5**, "Trained full") or training directly on sparsely-sampled data (**Fig. 5**, "Trained sparse", same as in **Fig. 2b**). These results show that dynamics models are learned most accurately on fully observed data, but that the learned dynamics can be used to model sparsely sampled data if models are adapted to the sparser domain using SBTT.

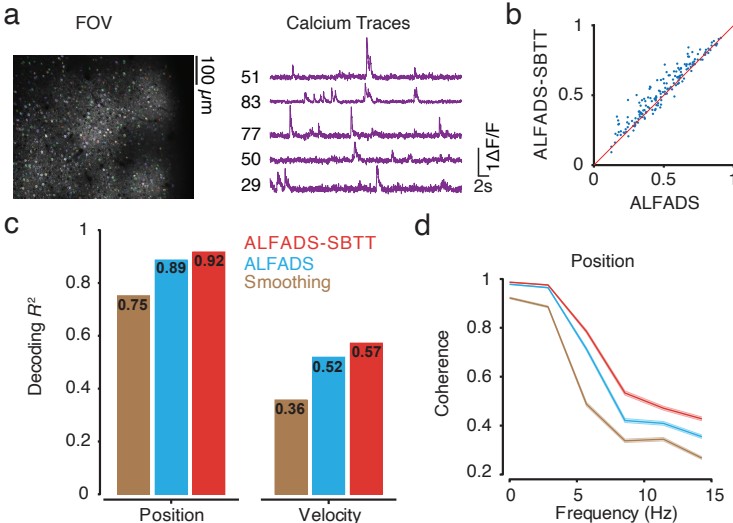

Figure 4: SBTT improves inference of latent dynamics from mouse 2P calcium imaging data. (a) Left: an example field-of-view (FOV), colored by neurons. Right: calcium traces (dF/F) from a single trial for 5 example neurons. (b) Performance of capturing empirical PSTHs was quantified by computing the correlation coefficient r between the inferred single-trial event rates and empirical PSTHs, comparing ALFADS vs ALFADS-SBTT. Each point represents an individual neuron. (c) Decoding performance was quantified by computing the $R^2$ between the true and decoded position (left) and velocity (right) across all trials. (d) Quality of reconstructing the kinematics across frequencies was quantified by measuring coherence between the true and decoded position for all three methods.

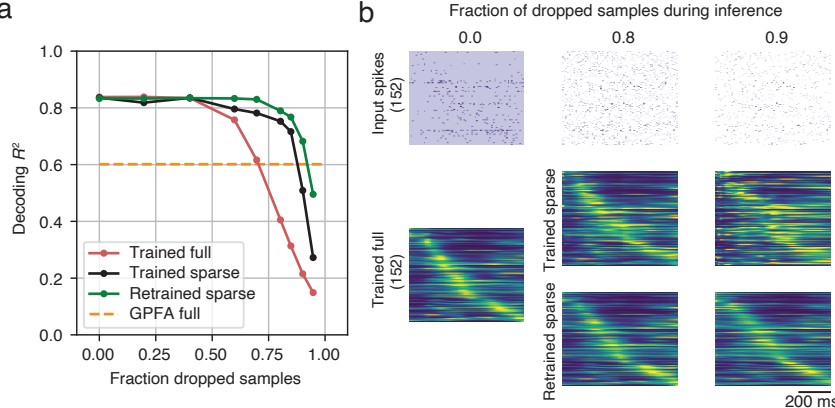

Figure 5: Retraining full-data LFADS encoders on sparse data improves decoding performance. (a) Hand velocity decoding performance in function of dropped samples (as in **Fig. 2b**). "Trained full" indicates training on fully observed data and inference on sparse data. "Trained sparse" indicates training and inference on sparse data. "Retrained sparse" indicates training on fully observed data, followed by encoder retraining and inference on sparse data. (b) Spike count input and inferred rate output of LFADS. Conventions are as in **Fig. 2a**.

# 5 Discussion

We introduced SBTT, a novel approach for learning latent dynamics from irregularly or sparsely sampled time series data. In experiments on real electrophysiology data from macaque motor cortex, we show that models trained with SBTT learn biologically relevant neural dynamics with up to 80% masked training data. On data from a synthetic 2P calcium imaging simulation, we show that models trained with SBTT capture high frequency features of the latent dynamics that are not captured at frame resolution. We also showed improved behavioral decoding performance on real 2P imaging data from mouse M1. Finally, we demonstrate that retraining the early layers of a full-data model on sparse datasets using SBTT can substantially improve decoding performance at the most challenging sparsity levels, outperforming models trained on the sparse data alone. Taken together, these results clearly show that SBTT is a valuable technique for training models with irregularly or sparsely sampled time series data.

## 5.1 Limitations

Though we made an effort to characterize performance across multiple potential applications, it remains untested how this approach would generalize to other experimental settings (microscopes, calcium indicators, expression levels), model systems, and brain areas or tasks with more complex or higher-dimensional dynamics [39], but we are optimistic that these properties will extend to AutoLFADS models that use SBTT in other settings. Applications to brain-machine interfaces await incorporation of neural network-based dynamics models into closed-loop, real time systems. We also note that hardware implementations of intermittent sampling for electrophysiology are still largely unexplored, and might incur time or power costs when switching between channels. This might change the point at which intermittent sampling is beneficial from a power or performance perspective. We hope that this work indicates new directions for future generations of recording hardware that focus on high interface capacities and rapid switching between contacts.

## 5.2 Broader impact

Our results could pave the way to substantially decreased power consumption for fully-implantable brain-machine interfaces. Ultimately, this should result in more reliable and less burdensome assistive devices for people with disabilities. Further, expanding the information that can be gathered through a given recording bandwidth has scientific implications, and could enable neuroscientists to ask new questions via larger-scale studies of the brain.

Like any resource-intensive technology, this technique has the potential to increase inequity by only benefiting those who can afford the most advanced neural interfaces. Efforts to deploy such technologies should weigh input from ethicists to ensure that everyone benefits from these scientific innovations [60, 61].

## Acknowledgments and Disclosure of Funding

We thank M. Rivers and R. Vescovi for help with the real-time camera setup, and D. Sabatini for contributions to the behavioral control software. This work was supported by the Emory Neuromodulation and Technology Innovation Center (ENTICe), NSF NCS 1835364, NIH Eunice Kennedy Shriver NICHD K12HD073945, the Simons Foundation as part of the Simons-Emory International Consortium on Motor Control (CP), the Alfred P. Sloan Foundation (CP, MTK), NSF Graduate Research Fellowship DGE-2039655 (ARS), NSF NCS 1835390, The University of Chicago, the Neuroscience Institute at The University of Chicago (MTK), and a Beckman Young Investigators Award (AG). The work was also supported by the following collaborative awards (PI: Prof. Ellen Hess, Emory): NIH NINDS R21 NS116311, Imagine, Innovate and Impact (I3) Funds from the Emory School of Medicine and through the Georgia CTSA NIH UL1-TR002378, and a pilot grant from the Emory Udall Center of Excellence for Parkinson's Research. The authors declare no competing interests.

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
