# Deep inference of latent dynamics with spatio-temporal super-resolution using selective backpropagation through time
## *Supplementary Material*

## A  Training the AutoLFADS models

### A.1  LFADS architecture

The architecture of LFADS is described in more detail in the original publication [1]. We used a dimension of 64 for the initial condition (IC) encoder, controller input (CI) encoder, initial condition, and controller. The controller output dimension was 2 and the generator dimension was 100. The latent factor dimensionality was 40 for the maze dataset and 100 for both calcium datasets.

### A.2  Hyperparameter tuning

LFADS models benefit from appropriate hyperparameter (HP) tuning, as optimal HP combinations can vary from dataset to dataset [2, 3]. As mentioned in the main text, we use AutoLFADS [3] to ensure appropriate HP tuning. The framework combines a regularization strategy (coordinated dropout; CD [2]) with a large-scale framework for optimizing model hyperparameters (population-based training; PBT [4]). The details of these strategies are outlined in previous work; here we provide specifics for replicating our findings.

PBT trains many models in parallel, while using evolutionary algorithms to exploit and explore high-performing HP combinations. In our experiments, we used 18 workers on a local cluster with 9 NVIDIA GeForce RTX 2080 GPUs (i.e., generations of 18 models were trained in parallel). The following HPs were searched using PBT, with initial values sampled from distributions listed in parentheses: Adam learning rate (0.001 for all models), CD rate (0.5 for all models), dropout rate ($Uniform(0, 0.7)$), L2 penalty on the generator recurrent weights ($LogUniform(-5, -1)$), L2 penalty on the controller recurrent weights ($LogUniform(-5, -1)$), KL penalty on the controller output ($LogUniform(-6, -4)$), and KL penalty on the initial condition distribution ($LogUniform(-6, -4)$). After an initial ramping period (80 epochs), during which regularization penalties were linearly increased to full strength, we trained all models in intervals of 50 epochs (one generation). For all the datasets used in this paper, trials were split into 80/20 training and validation sets. At the end of each generation, models were scored using exponentially smoothed negative log-likelihood (NLL) on validation data and participants in the next generation were selected with a binary tournament. Models with worse scores copied the weights and mutated HPs from the winning models. Training was stopped when a fractional improvement of less than 0.001 (maze data) or less than 0 (simulated calcium and real calcium data) in best-in-generation score was achieved over 25 generations. The model checkpoints at epochs with the best smoothed validation NLL were used for subsequent analysis.

## B  Zero-Inflated Gamma (ZIG) emissions model

Recent work demonstrated that deconvolved calcium events in 2P data can be robustly modeled with a zero-inflated gamma distribution [5]. We therefore replaced the Poisson emissions model of LFADS, which links latent factors to observed events, with the ZIG model. Concretely, a ZIG distribution is a two-component mixture model that combines a gamma distribution to model the continuous-valued deconvolved events and a point mass that represents the probability of zero events (missed spikes [5]):

$$y_n(t) \sim (1 - q_n(t)) \cdot \delta(0) + q_n(t) \cdot gamma(\alpha_n(t), k_n(t), loc_n),$$

where $y_n(t)$ is the distribution of observed deconvolved events, $\alpha_n(t)$ and $k_n(t)$ are the scale and shape parameters of the gamma distribution, and $q_n(t)$ denotes the probability of non-zeros, for neuron $n$ at time $t$. The location parameter $loc_n$ of the gamma distribution for neuron $n$ was fixed as the minimum nonzero deconvolved event ($s_{min}$) for that neuron. We modified LFADS so it infers the three time-varying parameters ($\alpha_n(t)$, $k_n(t)$, and $q_n(t)$) for each neuron. This is achieved through linear transformation of the factors followed by a trainable, scaled sigmoid nonlinearity. The outputs of the sigmoid for $\alpha_n(t)$ and $k_n(t)$ are scaled by positive parameters (one for each neuron) that are optimized alongside network weights. An L2 penalty is applied between the scaling factors and a PBT-searchable prior to prevent extreme values. The training objective is to minimize the negative log-likelihood of the deconvolved events given the inferred parameters:

$$\prod p(y_i(t)|\text{ZIG}(\hat{\alpha_i}(t), \hat{k_i}(t), \hat{q_i}(t)))$$

The event rate for neuron $n$ at time $t$ was estimated by taking the mean of the inferred ZIG distribution: $\hat{q_n}(t) \cdot (\hat{k_n}(t) \cdot \hat{\alpha_n}(t) + s_{min})$.

## C   Calcium simulations

### C.1   Simulation pipeline

Synthetic data were generated with underlying dynamics that follow a Lorenz system, as described in previous work [6, 7]. Lorenz parameters were set to standard values ($\sigma$: 10, $\rho$: 28, and $\beta$: 8/3) and $\Delta t$ was set to 0.01. We generated Lorenz systems with various speeds and frequency peaks by downsampling the original Lorenz states. We simulated a population of 278 neurons with firing rates taken as linear projections of the Lorenz state variables using random weights, followed by an exponential nonlinearity. Scaling factors were applied so that the baseline firing rate for all neurons was 3 spikes/sec. We simulated rates for 32 conditions and sampled spikes for 60 trials per condition. Each condition was obtained by starting the Lorenz system with a random initial state vector and running it for 900ms.

Generating realistic calcium traces from the synthetic spike trains followed a multi-step process. First, independent Gaussian noise ($s.d. = 0.1$) was added to each spike in the spike train to model the variability in spike amplitudes observed in real calcium data. Next, we modeled the calcium concentration dynamics ($c(t)$) as an autoregressive process of order 1:

$$c(t) = \gamma c(t - 1) + s(t) \tag{1}$$

with s(t) representing the number of spikes at time t and $\gamma \sim \mathbf{U}(0.93, 0.95)$ is the autoregressive coefficient uniformly distributed to account for variability across neurons in calcium imaging movies. Subsequently, we computed the noiseless fluorescence signals by passing the calcium dynamics through a nonlinear transformation estimated from the literature [8] for the calcium indicator GCaMP6f. After passing through a nonlinearity the relationship between spike size and trace size is corrupted, and therefore we rescaled the trace using min-max normalization. Finally, Gaussian noise ($\sim \mathbf{N}(0, sn)$) and Poisson noise (simulated as Gaussian with mean 0 and variance proportional to the signal amplitude at each time point via a constant $d$) were added to the normalized traces. The simulated fluorescence signals were deconvolved using OASIS parameterized with an order 1 auto-regressive model and $s_{min} = 0.1$ ($s_{min}$ corresponds to the location parameter of a ZIG distribution) [9].

The noise level associated with each fluorescence trace is a crucial parameter. High noise levels lead to very poor spike detection and very low noise levels enable a near-perfect reconstruction of the spike train. In order to select a fair level of noise we matched the SNR distributions of the simulated data to that of real data from motor cortex. SNR was estimated as the ratio between the noise level of the fluorescence

signal estimated by OASIS and the largest detected spike inferred by OASIS. We found that a truncated normal distribution of noise levels for Gaussian and Poisson noise best matched the SNR distributions. More precisely, for each neuron, $sn = d$ was sampled independently from a truncated normal distribution $\mathbf{N}(0.3, 0.02)$ truncated below 0.09. We also measured the correlation coefficient $r$ between the deconvolved events and ground truth spikes. With the above noise settings, the mean $r$ was 0.32, which is consistent with standard benchmarks [10] for OASIS. It is worth noting that real data feature a broad range of noise levels that depend on the imaging conditions, depth, expression level, laser power and other factors. In our setting the goal was not to investigate all possible noise conditions, but rather to provide simulated data whose properties roughly matched the features of the real calcium imaging data used in this paper.

## C.2 Mapping to ground truth Lorenz states

The A-ZIG-SBTT and A-ZIG-FR models output inferred calcium event rates at 100Hz and 33Hz respectively, whereas Gaussian smoothing outputs 33Hz smoothed deconvolved events. To evaluate the performance in recovering the 100Hz ground truth Lorenz states, the 33Hz outputs were linearly extrapolated to 100Hz.

Since our goal was to quantify modeling performance by estimating the underlying Lorenz States, we trained a mapping from the output of each model to the ground truth Lorenz states using ridge regression. First, we split the trials into training (80%) and test (20%) sets. We used the training set to optimize the regularization coefficient using 5-fold cross-validation, and used the optimal regularization coefficient to train the mapping on the full training set. We then quantified state estimation performance by applying this trained mapping to the test set and calculating the coefficient of determination ($R^2$) between the true and predicted Lorenz states. We repeated the above procedure five times with train/test splits drawn from the data in an interleaved fashion. We reported the mean $R^2$ across the repeats, such that all reported numbers reflect held-out performance.

The same cross-validated ridge regression procedure was used for the real calcium data, i.e., to train decoders to predict positions and velocities from the inferred event rates produced by each model.

# D   Real 2P Calcium imaging

We tested SBTT with real 2P calcium data that we collected from motor cortex in a mouse performing a forelimb water grab task. These data have not been published previously, and thus we provide detailed experimental methods below.

## D.1 Surgical procedures

All procedures were approved by the Animal Care and Use Committee at the institution where the experiments were performed. One male Ai148D transgenic mouse (TIT2L-GC6f-ICL-tTA2; Jackson Laboratory) was used and underwent a single surgery. The mouse was injected subcutaneously with dexamethasone (8 mg/kg) 24 hours and 1 hour before surgery. The mouse was anesthetized with 2-2.5% inhaled isoflurane gas, then injected intraperitoneally with a ketamine-medetomidine mixture (60 mg/kg ketamine, 0.25 mg/kg medetomidine), and maintained on a low level of supplemental isoflurane (0-1%) if it showed any signs that the depth of anesthesia was insufficient. Meloxicam was also administered subcutaneously (2 mg/kg) at the beginning of the surgery and for 1-3 subsequent days. The scalp was shaved, cleaned, and resected, the skull was cleaned and the wound margins glued to the skull with tissue glue (VetBond, 3M), and a 3 mm circular craniotomy was made with a 3 mm biopsy punch centered over the left CFA/S1 border. The coordinates for the center of CFA were taken to be 0.4 mm anterior and 1.6 mm lateral of bregma. Virus (AAV9-CaMKII-Cre, stock 2.1*1013 particles/nL, 1:1 dilution in PBS, Addgene) was pressure injected (NanoJect

III, Drummond Scientific) at multiple sites near the target site, with 140 nL injected at each of two depths per site (250 and 500 µm below the pia) over 5 minutes each. The craniotomy was then sealed with a custom cylindrical glass plug (3 mm diameter, 660 µm depth; Tower Optical) bonded (Norland Optical Adhesive 61, Norland) to a round coverslip and glued in place. A small craniotomy was also made using a dental drill over right CFA at 0.4 mm anterior and -1.6 mm lateral of bregma, where 140 nL of AAVretro-tdTomato (stock 1.02*1013 particles/nL, Addgene) was injected at 300 µm below the pia. This injection labeled cells in left CFA projecting to the contralateral cortex. Here, this labeling was used solely for stabilizing the imaging plane (see below). A custom laser-cut titanium head bar was affixed to the skull with black dental acrylic. The animal was allowed to recover at least 3 days before water restriction.

## D.2    Behavioral task

The water grab task was a variant of a previously-reported water reaching task [11]. This task was performed by a water-restricted, head-fixed mouse, with the forepaws beginning on metal paw rests and the hindpaws and body supported by an acrylic tube enclosure. After the mouse held the paw rests for 700-900 ms, a tone was played by stereo speakers and a droplet of water appeared at one of two water spouts positioned on either side of the snout. The tone's pitch indicated the location of the water, with a 4000 Hz tone indicating left and a 7000 Hz tone indicating right. The tone lasted 500 ms or until the mouse made contact with the correct water spout. The mouse could grab the water droplet and bring it to its mouth to drink any time after the tone began. Both the paw rests and spouts were wired with capacitive touch sensors (Teensy 3.2, PJRC). Good contact with the correct spout produced an inter-trial interval of 3-6 s, while failure to make contact (or insufficiently strong contact) with the spout produced an inter-trial interval of 20 s. Because the touch sensors required good contact from the paw, this setup encouraged complex contacts with the spouts. The mouse was trained to make all reaches with the right paw and to keep the left paw on the paw rest during reaching. Training took approximately two weeks, though the behavior continued to solidify for at least two more weeks. Data presented here were collected after 6-8 weeks' experience with the task. Touch event monitoring and task control were performed at 60 Hz.

Behavior was also recorded using a pair of cameras (BFS-U3-16S2M-CS, FLIR; varifocal lenses COZ2813CSIR2, Computar) mounted 150 mm from the right paw rest at 10 degrees apart to enable 3D triangulation. Infrared illuminators enabled behavioral imaging. Cameras were synchronized and recorded at 150 frames per second with real-time image cropping and JPEG compression, and streamed to one HDF5 file per camera. The knuckles and wrist of the reaching paw were tracked in each camera using DeepLabCut [12] and triangulated into 3D using camera calibration parameters obtained from the MATLAB Stereo Camera Calibration toolbox [13, 14]. To screen the tracked markers for quality we created distributions of all inter-marker distances in 3D across every labeled frame and identified frames with any inter-marker distance exceeding the 99.9th percentile of its respective distribution as problematic. Trials with more than one problematic frame in the period of -200 ms to 800 ms after the raw reach onset were discarded (where reach onset was taken as the first 60 Hz tick after the paw rest touch sensor fell below contact threshold). The kinematics of all trials that passed this screening procedure were visualized to confirm quality. Forepaw centroid marker kinematics were obtained by averaging the kinematics of all paw markers, locking them to behavioral events and then smoothing using a Gaussian filter (15 ms s.d.). To obtain velocity and acceleration, centroid data were numerically differentiated with MATLAB's `diff` function and then smoothed again using a Gaussian filter (15 ms s.d.).

## D.3    Two-photon imaging

Calcium imaging was performed with a Neurolabware two-photon microscope and pulsed Ti:sapphire laser (Vision II, Coherent). Depth stability of the imaging plane was maintained using a custom plugin that acquired an image stack at the beginning of the session (1.4 µm spacing), then compared a registered rolling average of the red-channel data to each plane of the stack. If sufficient evidence indicated that a different

plane was a better match to the image being acquired, the objective was automatically moved to compensate.

Offline, images were run through Suite2p [15] to perform motion correction, ROI detection, and fluorescence extraction from both ROIs and neuropil. ROIs were manually curated using the Suite2p GUI. We then subtracted the neuropil signal scaled by 0.7 [16]. Neuropil-subtracted ROI fluorescence was then detrended by performing a running 10th percentile operation, smoothing with a Gaussian (20s s.d.), then subtracting the result from the trace. This result was fed into OASIS [9] using the 'thresholded' method, AR1 event model, and limiting the tau parameter to be between 300 and 800 ms. Neurons were discarded if they did not meet a minimum signal-to-noise (SNR) criterion. To compute SNR, we took the fluorescence at each time point when OASIS identified an "event" (non-zero), computed (fluorescence - neuropil) / neuropil, and computed the median of the resulting distribution. ROIs were excluded if this value was less than 0.05. To put events on a more useful scaling, for each ROI we found the distribution of event sizes, smoothed the distribution (`ksdensity` in MATLAB, with an Epanechnikov kernel and log transform), found the peak of the smoothed distribution, and divided all event sizes by this value. This rescales the peak of the distribution to have a value of unity. Data from one mouse (one session) were used (439 neurons, 475 trials).

## D.4   Modeling with frame and sub-frame resolution

To prepare data for A-ZIG-SBTT and A-ZIG-FR, the deconvolved events were normalized by $s_{min}$ so that the minimal event size was 0.1 across all neurons. The deconvolved events for individual neurons had a sampling rate equal to the frame rate (31.08 Hz). For modeling with A-ZIG-SBTT, the deconvolved events were assigned into 10ms bins using the timing of individual measurements for each neuron to achieve sub-frame resolution (i.e., 100 Hz). For A-ZIG-FR and Gaussian smoothing, the deconvolved events were assigned into a single time bin per frame (i.e., 32.17 ms bins) to mimic standard processing of 2p imaging data, where the sub-frame timing of individual measurements is discarded. Trials were created by aligning the data to 200 ms before and 800 ms after reach onset (100 time points per trial for A-ZIG-SBTT, and 31 time points per trial for A-ZIG-FR and Gaussian smoothing). Failed trials (latency to contact with correct spout > 20 s), or trials where the grab to the incorrect spout occurred before the grab to the correct spout, were discarded.

## D.5   Evaluating against empirical PSTHs

The continuous range of reaching behavior was discretized into groups for trial-averaging. Trials were sorted into four groups based on the Z dimension of hand position. The hand position was obtained by smoothing the centroid marker position with a Gaussian filter (40 ms s.d.). Time windows where the hand Z position was used to split trials were selected arbitrarily to present a good separation between subgroups of hand trajectories. A window of 30 ms to 50 ms after reach onset was used to split both left and right condition trials. For both left or right conditions, 55 trials with the lowest and highest Z positions were selected as group 1 and group 2, respectively; trials with middle-range Z positions were discarded.

To assess how well the models' inferred event rates recapitulated the empirical PSTHs on single trials, empirical PSTHs were computed by trial-averaging smoothed deconvolved events (40 ms kernel s.d., 32.1729 ms bins) within each of the 4 subgroups of trials. Event rates inferred from A-ZIG-SBTT were first downsampled from 100 Hz to 31.0821 Hz with an antialiasing filter applied, to match the sampling frequency (i.e., the frame rate) of the original deconvolved signals. The correlation coefficient ($r$) was computed between inferred single-trial event rates and the corresponding empirical PSTHs for all active neurons for both methods (i.e., calculated on rates concatenated across all trials within the four subgroups; one $r$ for each neuron). Active neurons were defined as neurons that had more than 40 nonzero events across all trials from all 4 subgroups in the time window of 200 ms before to 800 ms after reach onset.

## D.6  Decoding hand kinematics

A-ZIG-SBTT inferred rates, A-ZIG-FR inferred rates, and smoothed deconvolved events (Gaussian kernel 40 ms s.d.) were used to decode hand position and velocity using ridge regression. The hand position and velocity were obtained as described above and binned at 10ms (i.e., 100 Hz). The non-A-ZIG-SBTT rates were retained to a sampling frequency of 100Hz using linear interpolation. For simplicity, we did not include a lag between the neural data and kinematics. However, additional analyses confirmed that adding a lag did not alter the results (data not shown). Trials with an interval between water presentation and reach onset that was longer than a threshold (i.e., 400ms) were discarded due to potential variations in behavior (e.g., inattention). The data were aligned to 50 ms before and 350 ms after reach onset. The decoder was trained and tested using the same cross-validated Ridge regression approach described in section C.2. The coefficient of determination ($R^2$) was computed and averaged across x-, y- and z- kinematics.

## D.7  Coherence analysis

Coherence was computed between the true and predicted kinematics (window: 200 ms before and 500 ms after reach onset) across all trials and across all x-, y- and z- dimensions using magnitude-squared coherence (MATLAB: `mscohere`). The power spectral density estimation parameters within `mscohere` were specified to ensure a robust calculation on the single trial activity. Hanning windows with 35 timesteps (i.e., 350 ms) for the FFT and window size, and 25 timesteps (i.e., 250ms) of overlap between windows. Coherence was evaluated at 18 frequencies that were evenly spaced between 0 Hz and half of the sampling frequency (i.e., 100Hz). The interval between the frequencies was determined by the window size passed to `mscohere`. In Fig. 4d, only coherence between 0 and 15Hz was plotted because coherence dropped low for all methods after 15Hz.

# E  Evaluation metrics

## E.1  Coefficient of determination

Decoding performance was quantified using the coefficient of determination ($R^2$) between true and predicted hand velocities as implemented in `sklearn` [17]. We compute $R^2$ for $x$- and $y$-dimensions separately, and then average.

$$R^2(y, \hat{y}) = 1 - \frac{\sum_{i=1}^{n}(y_i - \hat{y}_i)^2}{\sum_{i=1}^{n}(y_i - \bar{y})^2}$$

## E.2  Pseudo-$R^2$

The problem with evaluating our models using kinematic decoding alone is that it only gives information about the quality of the behaviorally relevant projections of the population state, while these could represent only a small fraction of the total variance of the population. To obtain a more complete picture of inference quality, we also want to characterize how well the inferred population state predicts the spiking activity of held-out neurons. Since spikes are Poisson-distributed, we fit a Poisson GLM (i.e., with exponential link function) to predict the firing rate of each held-out neuron based on the inferred population state. To quantify the performance of these models, we use pseudo-R2 as a likelihood-based metric similar to the coefficient of determination. We use an implementation from `pyglmnet` [18], where $ln\hat{L}$ is the estimated log-likelihood, $S$

is a model that predicts the spike counts, $M_{GLM}$ is the GLM, and $M_{null}$ is a model that predicts the mean count.

$$pR^2 = 1 - \frac{ln\hat{L}(S) - ln\hat{L}(M_{GLM})}{ln\hat{L}(S) - ln\hat{L}(M_{null})}$$