# OpenReview forum: "Deep inference of latent dynamics with spatio-temporal super-resolution using selective backpropagation through time"
_NeurIPS.cc/2021/Conference — NeurIPS 2021 Poster_

### Official Review · Reviewer_XQrZ · 2021-07-01

**Rating:** 5
**Confidence:** 4

**Summary:**

This paper proposes a correction to the error term used when optimizing inference algorithms for latent dynamics on data for which the available observations change over time.

Concretely, those missing terms are removed from the sum, i.e. zeroed out. The authors use this method to train LFADS models on a data set for behavioral decoding, showing that decoding performance only slightly deteriorates when working with a fraction of the full observations.

Furthermore, they apply the method to 2P calcium recordings, showing that accounting for the sub-bin timing leads to an increased decoding performance.

**Limitations And Societal Impact:**

Yes.

**Main Review:**

The paper is well written and clear. The experiments performed are reasonable. The performance gained on the 2P from using sub-bin timing information is an interesting result.

My main issue with the paper is that the proposed technique is somewhat apparent, and I wouldn't be surprised if it is already used by other people who deal with missing data points.

Therefore, while I believe the work is sound, I think it is not substantial enough to recommend acceptance.

Additional points:
It is not clear how the frequencies in Fig. 4d are determined

Some simple polishing of the figures regarding alignment and matching of fontsizes would considerably improve their visual appearance

**Time Spent Reviewing:**

3

---

> ### Author Response · Authors · 2021-08-10
> **Response to Reviewer XQrZ**
>
> **Summary:**
>
> >This paper proposes a correction to the error term used when optimizing inference algorithms for latent dynamics on data for which the available observations change over time.
>
> >Concretely, those missing terms are removed from the sum, i.e. zeroed out. The authors use this method to train LFADS models on a data set for behavioral decoding, showing that decoding performance only slightly deteriorates when working with a fraction of the full observations.
>
> >Furthermore, they apply the method to 2P calcium recordings, showing that accounting for the sub-bin timing leads to an increased decoding performance.
>
> **Main Review:**
>
> >The paper is well written and clear. The experiments performed are reasonable. The performance gained on the 2P from using sub-bin timing information is an interesting result.
>
> We appreciate the feedback, and are glad that the reviewer found the paper clear, the experiments reasonable, and the 2p result interesting.
>
> >My main issue with the paper is that the proposed technique is somewhat apparent, and I wouldn't be surprised if it is already used by other people who deal with missing data points.
>
> >Therefore, while I believe the work is sound, I think it is not substantial enough to recommend acceptance.
>
> We appreciate the feedback. Though the method is technically simple, there are two major innovations here. First, framing the limited temporal resolution of calcium imaging as a missing data problem is not obvious to those working with two-photon imaging data. Second, though SBTT may be a straightforward solution for missing data, we have found no direct precedent in the literature for this approach, and in combination with the dynamical systems approach of LFADS produces an extremely effective solution. If the reviewer knows of any closer work, this would be helpful information for us. We address the question of significance and novelty more fully in the above response to reviewer **EKdk**.
>
> >Additional points: It is not clear how the frequencies in Fig. 4d are determined.
>
> Apologies, we will provide more details in paper revision. We used `mscohere` in MATLAB to evaluate the coherence between the true and predicted kinematics. It evaluated coherence at 18 frequencies that were evenly spaced between 0 Hz and half of the sampling frequency (i.e., 100Hz). The interval between the frequencies was determined by the window length inputted to `mscohere`. In Fig. 4d, we plotted only coherence between 0 and 15Hz because coherence dropped low for all methods after 15Hz.
>
>
> >Some simple polishing of the figures regarding alignment and matching of font sizes would considerably improve their visual appearance.
>
> We will update figures for alignment and font sizes in the subsequent revision.

---

### Official Review · Reviewer_7Nfr · 2021-07-15

**Rating:** 4
**Confidence:** 5

**Summary:**

The authors propose a novel back-propagation method SBTT when partial data is missing. They mainly focus on latent dynamical models for neural population analysis and show extensive experiments comparing AutoLFADS with or w/o SBTT.


**Limitations And Societal Impact:**

The authors mention some competitive works such as coordinated dropout, masked language modeling, and DeepInterpolation, but have included none in the experiment to convince us that SBTT not only works with AutoLFADS but also significantly beats alternatives dealing with missing data. I believe latent dynamic discovery with missing data is not a new modeling topic. Papers, such as Niklas Gunnarsson et al 2021, Armand Comas Massagué et al NeurIPS 2020, all investigate similar topics inferring latent representations with temporal information from missing observation values. How does SBTT even compare with a simple imputation model? What if, instead of dropping the missing values, interpolating them? I appreciate the authors present extensive neuroscience experiments but the significance of the proposed method as a generalizable modeling tool is not convincing yet. It looks like incremental work when running AutoLFADS on missing neural data, given the dominating use of A-ZIG-SBTT but no other imputation or interpolation type of methods.


**Main Review:**

The SBTT idea is original, interesting, and practical given the good results. The paper is well written and easy to follow. But as a modeling paper that is supposed to make significant contributions to the community, I think the method is relatively simple and straightforward.


**Time Spent Reviewing:**

2 hours

---

> ### Author Response · Authors · 2021-08-10
> **Response to Reviewer 7Nfr (Part I)**
>
> **Summary:**
>
> >The authors propose a novel back-propagation method SBTT when partial data is missing. They mainly focus on latent dynamical models for neural population analysis and show extensive experiments comparing AutoLFADS with or w/o SBTT.
>
> **Main Review:**
>
> >The SBTT idea is original, interesting, and practical given the good results. The paper is well written and easy to follow. But as a modeling paper that is supposed to make significant contributions to the community, I think the method is relatively simple and straightforward.
>
> We thank the reviewer for recognizing that our work is original, interesting and practical for real neuroscience applications! We fully agree that the technical innovation of our method is intuitive and easy to implement. However, conceptually, we believe our work is novel and potentially highly impactful. We detailed the contribution and significance of our method in depth in the above response to reviewer **EKdk**. In addition, we believe the simplicity of the method is a strength, making it broadly applicable to a variety of modeling approaches. Additionally, reframing recording bandwidth limitations as missing data problems is a completely new idea in electrophysiology and calcium imaging and is a large part of the novelty of this work.
>
> **Limitations And Societal Impact:**
>
> >The authors mention some competitive works such as coordinated dropout, masked language modeling, and DeepInterpolation, but have included none in the experiment to convince us that SBTT not only works with AutoLFADS but also significantly beats alternatives dealing with missing data. I believe latent dynamic discovery with missing data is not a new modeling topic. Papers, such as Niklas Gunnarsson et al 2021, Armand Comas Massagué et al NeurIPS 2020, all investigate similar topics inferring latent representations with temporal information from missing observation values.
>
> We appreciate the suggestions of comparing some of the methods mentioned in the paper. We would like to first clarify a point of confusion. The approaches we mentioned in the paper - coordinated dropout, masked language modeling, and DeepInterpolation - are not competitive work to SBTT. These methods are related to SBTT in that they manipulate different subsets of data samples between network inputs and the loss computation. Thus, there are some conceptual similarities to the technical method by which SBTT is implemented. However, they do not address the problem that SBTT solves, which is to infer latent dynamics from incomplete data. We will revise the relevant paragraph to make this more clear.
>
> We also appreciate the reviewer bringing several recent works to our attention. Here, we provide our interpretation of their relevance.
>
> The authors of [1] use a VAE-Kalman filter to model the underlying dynamics of temporally distributed medical images (MRI, ultrasound, etc). A VAE is used to encode image frames such that these encodings are treated as observations in a linear state space Gaussian model. The parameters for both the VAE and the linear state space model are learned in an end to end manner. At inference time, a Kalman filter is used to generate smoothed/ filtered observation trajectories that a decoder can then use to generate image frames. The authors use this method to impute missing image frames. Most importantly, there is no mention on how this model is trained in the presence of missing observations. From what is provided in the paper, it appears that the network is trained using data that has no missing observations. Once the model is trained, the VAE-Kalman filter can be used to smooth/filter to impute missing observations. In contrast, our SBTT allows both training and evaluation on data with missing observations.
>
> The authors of [2] provide a method to impute objects that are obscured/missing in video frames. Their method requires prior knowledge about the number of objects in a video sequence. For each of these objects, a sequence of 3 latent variables - pose, appearance and missingness are inferred. If an object is inferred to be missing, a decoder estimates the pose and appearance of this object using a bidirectional LSTM. These estimated pose and appearance representations are used to generate the corresponding frame within the video sequence such that the missing object is correctly generated in the constructed video frame. Overall, this work is substantially different from ours for two main reasons. The first is that the authors have designed this solution specifically for high-dimensional videos and their network design choices are intended for videos (two of the latent representations variables are pose and appearance). On the other hand, our models naturally exploit latent structure in the neural data to infer missing points. The second is that the authors propose a solution that is much different from our work - we may simply ignore missing observations when computing gradients for back-propagation, while the authors of this work must first determine that an object within a video frame is missing and then generate a representation which is used to reconstruct the next frame.
>
> Our responses to the remaining reviews are in the next comment.
>
> [1] Gunnarsson, N., Sjölund, J., & Schön, T. B. (2021). Latent linear dynamics in spatiotemporal medical data. *arXiv preprint arXiv:2103.00930*.
> [2] Comas-Massagué, A., Zhang, C., Feric, Z., Camps, O., & Yu, R. (2020). Learning Disentangled Representations of Video with Missing Data. *arXiv preprint arXiv:2006.13391*.

---

> ### Author Response · Authors · 2021-08-10
> **Response to Reviewer 7Nfr (Part II)**
>
> >How does SBTT even compare with a simple imputation model? What if, instead of dropping the missing values, interpolating them? I appreciate the authors present extensive neuroscience experiments but the significance of the proposed method as a generalizable modeling tool is not convincing yet. It looks like incremental work when running AutoLFADS on missing neural data, given the dominating use of A-ZIG-SBTT but no other imputation or interpolation type of methods.
>
> We appreciate the suggestions; the issues here are subtle. We should first highlight one of the advantages of LFADS+SBTT, which we didn’t explicitly mention in the paper, which is that it infers the dynamics at both observed and missing timesteps while de-noising the data at the same time. This is important because it allows us to bypass the step of imputing or interpolating the noisy data at the beginning. For example, spike train data can be modeled as generated by a point process. Standard imputation methods, such as mean or median imputation, or interpolation methods, such as linear or spline interpolation, are grossly mismatched to this type of data. Moreover, because the spike train reflects the underlying time-varying firing rate of the neuron, imputation by a constant mean or median would eliminate the temporal structure driven by the underlying dynamics. Any model-based imputation/interpolation has to assume an underlying model, which is not known as a prior in our applications. For spike trains, common choices would be smoothness of the rates or an autoregressive process. Here, we take this a large step further by utilizing the inferred underlying dynamics, which we can do naturally in LFADS because the model learns exactly those dynamics. For calcium data, there is the further challenge of trying to infer spike trains from sparsely sampled fluorescence transients, which we tackle with deconvolution  However, sub-frame imputation is not efficient with deconvolution alone, because both the amplitude and exact timing of events is far too noisy.
>
> Although interpolation of the inputs is fraught without SBTT, we have tested interpolation of the outputs in our comparisons, which we would like to highlight here (Fig. 3 & 4; Supplemental sections C.2 & D.6). Instead of imputing or interpolating the input data, we interpolated the inferred rates from a frame rate of 33Hz (simulation) or 31Hz (mouse experiment) to 100Hz in our calcium experiments. The inferred rates are continuous and smooth, making them more suitable for interpolation than the raw spike train or calcium events. It is worth noting that the interpolation also incorporated the sub-frame phase information, as described in responses to reviewer **bwjY**. Yet, SBTT still outperforms these approaches, highlighting the importance of utilizing latent dynamics to infer the missing points.
>
> We also think adding a comparison with a state-of-the-art method would be a powerful addition to our paper, as it can show the generalizability of our method and also provide evidence for why AutoLFADS is a particularly good model to apply SBTT on. We note that our strategy of training neural networks with missing data can be adapted to other latent variable models as mentioned in our paper. However, we wanted to be precise about naming because applying the strategy to networks that do not explicitly model dynamics does not involve backpropagation “through time”. Therefore, when we apply the same approach to such networks we will refer to it as selective backpropagation (SB).
>
> With the motivation described above, we performed additional experiments with transformers, namely Neural Data Transformers (NDT) [3]. This recently-developed approach was tested on the same monkey M1/PMd dataset used in Section 4.1, so we did not have to go through the tough task of adapting a new ML model for neural data and find appropriate hyperparameter ranges for the given dataset (which is a substantial effort). Moreover, NDT performance nearly matched AutoLFADS on these data, meaning it is an alternate state-of-the-art method for neural data. NDT uses a stack of transformer layers that process the input data (i.e., the NDT encoder) whose outputs are linearly transformed and exponentiated to obtain the inferred firing rates. The architecture of NDT is not sequential and it does not use variational inference [3]. Therefore, applying SB to NDT (NDT+SB) provides a way of testing another state-of-the-art method, while also allowing us to test the generality of our concept beyond VAE architectures.
>
> Table 1 shows the decoding performance of NDT and AutoLFADS with increasing fractions of dropped samples. The task, data and decoding method are the same as detailed in Section 4.1 of our paper. Both NDT and AutoLFADS achieve high performance on fully observed data, with AutoLFADS slightly outperforming NDT. The decoding performance of NDT remains high when the fraction of dropped samples is ≤ 0.4 and is about the same as GPFA applied to fully sampled data at a fraction of 0.6 dropped, suggesting that SB can be beneficial for NDT. However, the rate at which the decoding performance declines with increasing fractions of dropped samples is worse for NDT than AutoLFADS.
>
> A key difference between NDT and AutoLFADS is that AutoLFADS has an explicit dynamics model, whereas NDT does not. Thus our new results suggest two key points: First, SBTT (or SB) can be scaled to one of the other architectures (NDT). Second, the concept used in SBTT is more effective when applied to methods that have a latent dynamics model (e.g., AutoLFADS) than other state-of-the-art methods that do not model explicit dynamics (e.g., SB on NDT). In the paper revision, we will show the recovered samples learned by NDT as requested, and combine these NDT results with current Figure 2.
>
> | Fraction of dropped samples | NDT+SB | AutoLFADS+SBTT |
> | :---: | :---: | :---: |
> | 0 | 0.82 | 0.89 |
> | 0.2 | 0.80 | 0.88 |
> | 0.4 | 0.74 | 0.89 |
> | 0.6 | 0.60 | 0.86 |
> | 0.7 | 0.49 | 0.84 |
> | 0.8 | 0.28 | 0.80 |
> | 0.85 | N/A | 0.75 |
> | 0.9 | N/A | 0.55 |
> | 0.95 | N/A | 0.29 |
>
> Table 1. Comparison of decoding performance on monkey M1/PMd electrophysiological data between NDT and AutoLFADS with increasing fractions of dropped samples (NDT+SB not run for fraction of dropped samples >= 0.85). Decoding accuracy was quantified by measuring variance explained ($R^2$) between the true and decoded hand velocity.
>
> Finally, we appreciate that the reviewer recognizes our extensive demonstration with extensive neuroscience experiments, and we believe this demonstrates the potential broad applications of SBTT for the neuroscience community.
>
> [3] Ye, J., & Pandarinath, C. (2021). Representation learning for neural population activity with Neural Data Transformers. *bioRxiv*.

---

### Official Review · Reviewer_bwjY · 2021-07-16

**Rating:** 4
**Confidence:** 4

**Summary:**

This paper addresses the problem of bandwidth limits of BCIs, and proposes to obtain spatio-temporal super-resolution data by inferring missing samples of neural data with a latent low-dimensional dynamics. The main contribution of this paper lies in that it proposes a selective backpropagation through time (SBTT) algorithm for network learning with missing samples, and applies to two different neural datasets of spike trains and calcium imaging data.

**Ethical Concerns:**

No.

**Limitations And Societal Impact:**

Yes.

**Main Review:**

1. The core idea of SBTT is to add a dropout mask in reconstruction error computation to indicate which data samples are missing, such that they do not corrupt the gradient signals. The idea is actually not novel in machine learning area, and an example is given in [1]. The sequential variational auto-encoder model is from a previous work of [36] in the paper. So I think the innovation of this paper is a combination of the two sides, which is not significant.

2. The main limitation lies in that, in neural spike signals, we usually do not know which are the missing samples, so that the mask is impossible to get in real data. For example, in spike trains, if a sample is 0, how can we discriminate whether it is a missing spike or the neuron is just not firing?

3. In the experiment of neural spike decoding, only a Gaussian process factor analysis (GPFA) method is compared. Decoding performance of Kalman filter with different fraction of dropped samples should be given, to see how much the missing samples influence the decoding performance, which can help support the contribution of missing data inference.

4. Can you specific how to calculate the pR2? Why pR2 is chosen for evaluation and comparison?

5. In the experiment of calcium imaging data, the input for SBTT contains sub-frame phase information, while only collapsed samples without sub-frame phase information are input to the approaches in comparison. Can the difference in input data lead to different results? Because sub-frame phase information can also be useful for decoding.


[1] Yoon, Jinsung, James Jordon, and Mihaela Schaar. "Gain: Missing data imputation using generative adversarial nets." International Conference on Machine Learning. PMLR, 2018.


**Time Spent Reviewing:**

4

---

> ### Author Response · Authors · 2021-08-10
> **Response to Reviewer bwjY**
>
> >**Summary:**
>
> >This paper addresses the problem of bandwidth limits of BCIs, and proposes to obtain spatio-temporal super-resolution data by inferring missing samples of neural data with a latent low-dimensional dynamics. The main contribution of this paper lies in that it proposes a selective backpropagation through time (SBTT) algorithm for network learning with missing samples, and applies to two different neural datasets of spike trains and calcium imaging data.
>
> Though BCIs are a significant target application of this work, we’d like to emphasize that this work has substantial potential to impact the way data is efficiently collected and modeled in many areas of neuroscience, especially as the field trends toward ultra large-scale and multi-area recordings that aim to provide a more complete picture of brain activity. Furthermore, our paper addresses a fundamental problem with the standard way of processing 2-photon (2p) calcium imaging data (a widely-used method in neuroscience), which is that the temporal resolution is limited by the slow frame rate. Our work provides a way to link subframe time information to population dynamics, which substantially increases the temporal resolution of the inference of latent dynamics, and thus the value of this workhorse method.
>
> >**Main Review:**
>
> >The core idea of SBTT is to add a dropout mask in reconstruction error computation to indicate which data samples are missing, such that they do not corrupt the gradient signals. The idea is actually not novel in machine learning area, and an example is given in [1]. The sequential variational auto-encoder model is from a previous work of [36] in the paper. So I think the innovation of this paper is a combination of the two sides, which is not significant.
>
> We appreciate the reviewer drawing our attention to related work. The reviewer is correct that this work builds on state-of-the-art models that have been previously published, which we cite in our manuscript. However, it is incorrect to say that the reference provided invalidates the novelty of this work, as it has a fundamentally different mechanism of action and does not model underlying dynamics. In the reference provided [1], Yoon et. al. also utilize a mask to identify indices where observations are missing. However, the manner in which this mask is actually used is very different from what we propose. Specifically, Yoon et al. use this mask to locate indices for missing observations, then use a GAN to impute these missing data by learning the distribution of the available observations through a discriminator network. In our work, we use the mask to ignore missing observations when computing the gradient for back propagation, and the inferred factors are produced through an RNN dynamical system. This, we believe, works so well because neural populations in many brain areas, including the ones we show in this paper (M1/PMd in monkey and M1 in mouse) act as a relatively low-dimensional dynamical system. Yoon et al.’s work is not designed to take advantage of these spatio-temporal relationships across time.
>
> We further address the question of significance and novelty in the above response to reviewer **EKdk**.
>
> >The main limitation lies in that, in neural spike signals, we usually do not know which are the missing samples, so that the mask is impossible to get in real data. For example, in spike trains, if a sample is 0, how can we discriminate whether it is a missing spike or the neuron is just not firing?
>
> We believe we should clarify an important point about our work. We do not treat an observation of 0 spikes as missing data -- on the contrary, a known lack of spikes is an informative observation. Instead, we treat unsampled time points as missing data. In 2p imaging, the missing data occur when the laser spot is not pointed at a given neuron, and this is known because the experimenter built the control system for the microscope’s mirrors. In electrophysiology, the recording capacity of the interface is switched between channels, and the recorded channels at any given time point are similarly controlled by the experimenter and therefore known.
>
> >In the experiment of neural spike decoding, only a Gaussian process factor analysis (GPFA) method is compared. Decoding performance of Kalman filter with different fraction of dropped samples should be given, to see how much the missing samples influence the decoding performance, which can help support the contribution of missing data inference.
>
> Thanks for the suggestion. First, we’d like to clarify why we included a comparison point in the experiment of neural spike decoding. The primary goal of this paper is to introduce a strategy to train neural networks on incomplete data driven by latent dynamics. Our objective is to show how this innovation can address practical problems in broad neuroscience applications, but not to compare different strategies to deal with missing data. The goal of comparing to GPFA with fully observed data, which serves as a standard benchmark in the field (see responses to reviewer **EKdk**), is to highlight that dropping e.g., 85% of samples with AutoLFADS+SBTT still gives quite good performance. While we agree that the Kalman filter can certainly be applied in settings with missing data using the Expectation-Maximization (EM) algorithm, a deep dive into comparing to non-neural network methods seems a distraction from the goal of this paper.
>
> Second, in seeking high performance inference it is well-established that LFADS outperforms linear dynamical system (LDS) models similar to the Kalman filter (e.g., PfLDS in [2]). It is worth noting that LDS models have similar performance with GPFA [2], but GPFA is more widely used in the neuroscience community. Since our goal here is to include a comparison point that the neuroscience community is more familiar with, we chose GPFA.
>
> We further justify our use of GPFA in the above response to reviewer **EKdk**.
>
> >Can you specific how to calculate the pR2? Why pR2 is chosen for evaluation and comparison?
>
> The problem with evaluating our models using kinematic decoding alone is that it only gives information about the quality of the behaviorally relevant projections of the population state, while these could represent only a small fraction of the total variance of the population. To get a more complete picture of inference quality, we also want to characterize how well the inferred population state predicts the spiking activity of held-out neurons. Since spikes are Poisson-distributed, we fit a Poisson GLM (i.e., with exponential link function) to predict the firing rate of each held-out neuron based on the inferred population state. To quantify the performance of these models, we use pseudo-R2 as a likelihood-based metric similar to the coefficient of determination. We use the following [implementation](https://github.com/glm-tools/pyglmnet) from `pyglmnet` [3], where $ln\hat{L}$ is the estimated log-likelihood, $S$ is a model that predicts the spike counts, $M_{GLM}$ is the GLM, and $M_{null}$ is a model that predicts the mean count.
>
> $$pR^2 = 1 - \frac{ln \hat{L}(S) - ln \hat{L}(M_{GLM})}{ln \hat{L}(S) - ln \hat{L}(M_{null})}$$
>
> We’ll be sure to add this reference to the paper.
>
> >In the experiment of calcium imaging data, the input for SBTT contains sub-frame phase information, while only collapsed samples without sub-frame phase information are input to the approaches in comparison. Can the difference in input data lead to different results? Because sub-frame phase information can also be useful for decoding.
>
> This is a great question. Taking the sub-frame phase information into account is exactly what SBTT enables. The reviewer is certainly correct that discarding the sub-frame phase information is suboptimal when linking the neural activity to behavior, but this is exactly the point of comparison here - incorporating sub-frame timing in the inputs has not previously been addressed, and SBTT enables this when extracting the latent dynamics. To perform this comparison as fairly as possible, we tried upsampling the outputs: for A-ZIG-FR and Smoothing we interpolated the 31 Hz (collapsed) inferred rates (the outputs) up to 100 Hz rates using the sub-frame phase information (Fig. 3 & 4; Supplemental sections C.2 & D.6). SBTT outperformed this approach, highlighting the importance of keeping the sub-frame timing in the inputs used to infer the latent dynamics. We should also note that there have been efforts in utilizing sub-frame phase information for more precisely linking neural activity to behavior [4, 5]. However, these methods either require prior knowledge of the behavior or require stereotyped behavior across trials. SBTT does not require these constraints and is the first demonstration of linking sub-frame timing to latent dynamics.
>
> [1] Yoon, Jinsung, James Jordon, and Mihaela Schaar. "Gain: Missing data imputation using generative adversarial nets." International Conference on Machine Learning. PMLR, 2018.
>
> [2] Sussillo, D., Jozefowicz, R., Abbott, L. F., & Pandarinath, C. (2016). LFADS-Latent Factor Analysis via Dynamical Systems. arXiv preprint arXiv:1608.06315.
>
> [3] Jas, M., Achakulvisut, T., Idrizović, A., Acuna, D., Antalek, M., Marques, V., ... & Ramkumar, P. (2020). Pyglmnet: Python implementation of elastic-net regularized generalized linear models. Journal of Open Source Software, 5(47).
>
> [4] Picardo, M. A., Merel, J., Katlowitz, K. A., Vallentin, D., Okobi, D. E., Benezra, S. E., ... & Long, M. A. (2016). Population-level representation of a temporal sequence underlying song production in the zebra finch. Neuron, 90(4), 866-876.
>
> [5] Mano, O., Creamer, M. S., Matulis, C. A., Salazar-Gatzimas, E., Chen, J., Zavatone-Veth, J. A., & Clark, D. A. (2019). Using slow frame rate imaging to extract fast receptive fields. Nature communications, 10(1), 1-13.

---

### Official Review · Reviewer_EKdk · 2021-07-23

**Rating:** 5
**Confidence:** 4

**Summary:**

The authors propose a method called "selective back propagation through time" to estimate the latent dynamics of population activity for an intermittently sampled population of neurons. Of particular importance are the scenarios where large-scale electrophysiological recordings toggle through electrodes which are active in a given time window and 2-photon calcium imaging, which rasters over an imagining plane over relatively large time scales relative to neural dynamics.  The claim is that the method can circumvent the sampling rate of rastering and estimate dynamics with "super-resolution" by learning the underlying dynamics of the population with an explicit model for the rasterized sampling. The underlying dynamics are modeling using LFADS, an existing modeling framework and they compare their method to GPFA and to LFADS without accounting for the rasterized sampling. The models are compared using both synthetic and real data experiments and demonstrate superior performance to the proposed alternatives.

**Limitations And Societal Impact:**

yes.

**Main Review:**

I found this a very enjoyable and easy paper to read and I think it's addressing an important problem. In particular, this paper represents a contribution at the intersection of recording hardware, signal processing, and machine learning that is becoming increasingly important at very large recording scales. However, I have 2 big reservations about scoring this paper as highly as I would have liked to.

First, the main methodological contribution (SBTT) itself appears to be a fairly obvious and simple extension of existing methods and is reminiscent of (and somewhat less technically challenging than) the "stitching" method proposed in the original LFADS paper referenced in the present submission (although I may be missing something about the technical contribution here which should be expressed more explicitly). It's not clear to me what other contributions to consider that would warrant a stand-alone publication.

Second, the benchmark method compared against is a woefully insufficient straw man. In particular, the authors use GPFA as a comparison model but this is not apples-to-apples. At the very least the authors should be comparing against Poisson-GPFA so that the observation models are comparable.
3 related notes about these comparisons:
- the authors do not indicate the number of latent factors used for the LFADS model so it is difficult to assess the comparison with GPFA with 40 latent factors (line 190).
- Section 4.1 - The sampling rate of LFADS is higher than the sampling rate for GPFA. Why? Shouldn't they be the same in order to make a fair comparison?
- It would also be nice to see a side-by-side comparison of the computational resources required for the different models used as I believe this is an important factor in determining whether this is a feasible solution for a given lab.

Minor points:
- Line 221 - The sampling rate of the Lorenz model neurons (100Hz) seems unrealistically high.
- Section 4.3 - The authors use coherence as a measure of decoding accuracy. Why? Is there something rhythmic about the data?

I would add that I'm very open to increasing my score for this paper should the authors address these concerns sufficiently.

**Time Spent Reviewing:**

3

---

> ### Author Response · Authors · 2021-08-10
> **Response to Reviewer EKdk (Part I)**
>
> >Summary:
>
> >The authors propose a method called "selective back propagation through time" to estimate the latent dynamics of population activity for an intermittently sampled population of neurons. Of particular importance are the scenarios where large-scale electrophysiological recordings toggle through electrodes which are active in a given time window and 2-photon calcium imaging, which rasters over an imagining plane over relatively large time scales relative to neural dynamics. The claim is that the method can circumvent the sampling rate of rastering and estimate dynamics with "super-resolution" by learning the underlying dynamics of the population with an explicit model for the rasterized sampling. The underlying dynamics are modeling using LFADS, an existing modeling framework and they compare their method to GPFA and to LFADS without accounting for the rasterized sampling. The models are compared using both synthetic and real data experiments and demonstrate superior performance to the proposed alternatives.
>
> >Main Review:
>
> >I found this a very enjoyable and easy paper to read and I think it's addressing an important problem. In particular, this paper represents a contribution at the intersection of recording hardware, signal processing, and machine learning that is becoming increasingly important at very large recording scales.
>
> Thank you for the summary. We appreciate the comments, and are glad that the reviewer found the paper enjoyable and the problem significant.
>
> >However, I have 2 big reservations about scoring this paper as highly as I would have liked to.
>
> >First, the main methodological contribution (SBTT) itself appears to be a fairly obvious and simple extension of existing methods and is reminiscent of (and somewhat less technically challenging than) the "stitching" method proposed in the original LFADS paper referenced in the present submission (although I may be missing something about the technical contribution here which should be expressed more explicitly). It's not clear to me what other contributions to consider that would warrant a stand-alone publication.
>
> To briefly clarify a possible point of confusion, the stitching method in the original LFADS paper is used for incorporating non-simultaneously-collected data - sampled across different recording sessions (and hence from different sets of recorded neurons) - into the same dynamics model. This addresses a fundamentally different problem than SBTT.
>
> We fully agree that our work is not a complex algorithmic innovation over previous frameworks to infer latent dynamics. However, conceptually and practically, the work is novel and potentially highly impactful. While SBTT may be intuitive and an easy modification of existing frameworks - e.g. AutoLFADS [1], or LDS models that rely on ‘black box’ variational inference (PfLDS [2], GCLDS [3]) - our innovation is critically enabling for adapting this class of approaches to a new application domain. We also feel that the simple and intuitive solutions can nonetheless be very valuable, especially when thoroughly validated on multiple datasets. And the inherent simplicity of SBTT also lends generality, which could allow developers to easily adapt our solution to alternate models.
> We also feel that there is also a broader conceptual contribution: highlighting the growing space-time tradeoff in neural interfacing, the novel casting of it as a missing data problem, and presenting a solution in the form of latent dynamical models. Further, demonstrating that missing large amounts of data is not a deal-breaker may inspire innovation in casting other problems this way in multiple domains. We hope it motivates a larger design space for electrophysiological interfaces, e.g., inspiring interfaces that allocate their limited bandwidth toward infrequent sampling from much larger numbers of channels, rather than frequent sampling from a limited number. We also hope the demonstration inspires new sampling strategies in 2P imaging, such as using high-speed sampling of each of several, limited fields of view, before switching to interleaved sampling of all of those FOVs.
>
> [1] Keshtkaran, M. R., Sedler, A. R., Chowdhury, R. H., Tandon, R., Basrai, D., Nguyen, S. L., ... & Pandarinath, C. (2021). A large-scale neural network training framework for generalized estimation of single-trial population dynamics. bioRxiv.
>
> [2] Gao, Y., Archer, E. W., Paninski, L., & Cunningham, J. P. (2016). Linear dynamical neural population models through nonlinear embeddings. Advances in neural information processing systems, 29, 163-171.
>
> [3] Gao, Y., Busing, L., Shenoy, K. V., & Cunningham, J. P. (2015). High-dimensional neural spike train analysis with generalized count linear dynamical systems. Advances in neural information processing systems, 28, 2044-2052.

---

> ### Author Response · Authors · 2021-08-10
> **Response to Reviewer EKdk (Part II)**
>
> >Second, the benchmark method compared against is a woefully insufficient straw man. In particular, the authors use GPFA as a comparison model but this is not apples-to-apples. At the very least the authors should be comparing against Poisson-GPFA so that the observation models are comparable. 3 related notes about these comparisons:
>
> We use GPFA as a comparison model in the experiments on maze data for several reasons. First, GPFA is a standard benchmark in the field - it is used frequently on spike count data in the neuroscience community and provides readers in the field with a recognizable baseline. The Gaussian observation model was used on neural spiking data in the original publication [4] and was used as a baseline in the original LFADS paper [5]. We contacted the creators of GPFA, who tried a Poisson observation model when developing the method. They informed us that while Poisson is a more accurate observation model for spiking, it led to unstable learning and was computationally expensive, negating many of the practical benefits of GPFA. Second, to our knowledge, no publication has demonstrated a substantial gain of Poisson-GPFA over GPFA. For example, in previous comparisons between LFADS, GPFA, and vLGP (which is closely related to Poisson-GPFA), GPFA outperformed vLGP, and LFADS substantially outperformed both [6]. Like Poisson-GPFA, vLGP is computationally more expensive than GPFA and requires substantial hyperparameter tuning. In contrast, using standard GPFA ensures that the performance achieved is unlikely to be limited by hyperparameter optimization. Finally and most importantly, the goal of this analysis is not to rehash the superiority of LFADS over GPFA, but to show that LFADS with SBTT is extremely tolerant of missing data, maintaining quite good performance relative to a common, full-data baseline even when 85% of the samples have been dropped. Ultimately, we acknowledge that there are multiple potential points of reference; we chose a widely-used one (GPFA) as a “standard” baseline for comparison. Given its widespread use we would argue it is not a straw man, and we have not seen evidence that a Poisson-GPFA approach would perform substantially better.
>
>
>
> > - the authors do not indicate the number of latent factors used for the LFADS model so it is difficult to assess the comparison with GPFA with 40 latent factors (line 190).
>
> Thanks, this was an omission. We used the same number of latent factors for both models (40). We will make this explicit in the revision.
>
> > - Section 4.1 - The sampling rate of LFADS is higher than the sampling rate for GPFA. Why? Shouldn't they be the same in order to make a fair comparison?
>
> Thank you for this comment. We realize we did not make this point clear. Previous work has shown that 20 ms is the optimal bin size for GPFA when applied to the same dataset we use in this paper, as measured by decoding performance (supplementary figure 4 in [5]). Thus, we consider GPFA with 20 ms bins to be a well-established baseline for this dataset and therefore adopt it for our paper. We will clarify this point in the manuscript. On the other hand, AutoLFADS is not as sensitive to bin size and smaller bins allow us to demonstrate inference at a finer resolution. We have now confirmed that AutoLFADS factors from models trained on the full data achieve comparable velocity decoding at 10 and 20 ms bins ($R^2$ of 0.84 and 0.83, respectively).
>
> > - It would also be nice to see a side-by-side comparison of the computational resources required for the different models used as I believe this is an important factor in determining whether this is a feasible solution for a given lab.
>
> This is a great point. SBTT does not change the computational complexity of the model it is used with, but cost is an important consideration. The original AutoLFADS paper [7] includes this information, and we will note this.
>
> >Minor points:
>
> > - Line 221 - The sampling rate of the Lorenz model neurons (100Hz) seems unrealistically high.
>
> Sorry, this was not clear in the manuscript. The mentioned 100 Hz is the sampling frequency to simulate the events underlying the synthetic calcium data; it is not the simulated imaging rate or the firing rate of the neurons. This step was followed by the addition of noise processes relevant to calcium imaging, as well as downsampling to a 33 Hz frame rate, which is more realistic for a calcium imaging scenario. In case the reviewer is suggesting that our frequency of sampling is too high relative to the speed of the Lorenz dynamics, we note that without SBTT, the models we tested are unable to pick up the higher frequency oscillations even when the inferred rates are interpolated to 100 Hz. We will revise the manuscript to clarify this point, and are also happy to discuss if it is still unclear.
>
> > - Section 4.3 - The authors use coherence as a measure of decoding accuracy. Why? Is there something rhythmic about the data?
>
> Thanks. To clarify, coherence analysis is a useful method for determining how strongly two time series relate in different frequency bands [8]. In our application, we used coherence to compare the mouse’s true hand trajectories with the hand trajectories decoded from each method on a frequency-by-frequency basis. This is a more in-depth analysis than the high-level decoding summary (bar plots in Fig. 4c); the decoding summary simply tells us that SBTT allows ALFADS to achieve higher decoding performance, while coherence analysis tells us that the largest differences in decoding performance with SBTT (versus ALFADS with standard frame rates) were at high frequencies. This analysis is unrelated to whether the data are rhythmic or not.
>
> >I would add that I'm very open to increasing my score for this paper should the authors address these concerns sufficiently.
>
> We appreciate your feedback and willingness to discuss the manuscript!
>
>
> [4] Byron, M. Y., Cunningham, J. P., Santhanam, G., Ryu, S. I., Shenoy, K. V., & Sahani, M. (2009). Gaussian-process factor analysis for low-dimensional single-trial analysis of neural population activity. In Advances in Neural Information Processing Systems (pp. 1881-1888).
>
> [5] Pandarinath, C., O’Shea, D. J., Collins, J., Jozefowicz, R., Stavisky, S. D., Kao, J. C., ... & Sussillo, D. (2018). Inferring single-trial neural population dynamics using sequential auto-encoders. Nature Methods, 15 (10), 805-815.
>
> [6] Sussillo, D., Jozefowicz, R., Abbott, L. F., & Pandarinath, C. (2016). LFADS-Latent Factor Analysis via Dynamical Systems. arXiv preprint arXiv:1608.06315.
>
> [7] Keshtkaran, M. R., Sedler, A. R., Chowdhury, R. H., Tandon, R., Basrai, D., Nguyen, S. L., ... & Pandarinath, C. (2021). A large-scale neural network training framework for generalized estimation of single-trial population dynamics. bioRxiv.
>
> [8] Bokil, H., Andrews, P., Kulkarni, J. E., Mehta, S., & Mitra, P. P. (2010). Chronux: a platform for analyzing neural signals. Journal of Neuroscience Methods, 192(1), 146-151.

---

### Author Response · Authors · 2021-08-10
**Response Summary**

Thanks for all of this helpful feedback. Overall, reviewers noted that our method was relevant, interesting, and practical. They recognized the utility of this technique in neuroscience and appreciated our efforts to demonstrate on several datasets.

Reviewer **EKdk** pointed out that “*this paper represents a contribution at the intersection of recording hardware, signal processing, and machine learning that is becoming increasingly important at very large recording scales*”; reviewer **7Nfr** notes that that “*The SBTT idea is original, interesting, and practical given the good results*”; and reviewer **XQrZ** states that “*The performance gained on the 2P from using sub-bin timing information is an interesting result*”. Reviewers were also positive about the clarity and organization of the paper. Reviewer **EKdk** said ours was "*a very enjoyable and easy paper to read*" and thought it was "*addressing an important problem*.”

The reviewers’ concerns fell into two main categories: (i) technical novelty, with the concern that the simplicity of the technique was a weakness; and (ii) that SBTT needed further comparison and validation against other methods.

We’d like to make a few high-level points to address these concerns. First, we looked into all of the references provided. These methods either solved different problems, or had fundamentally different mechanisms of action. Detailed commentary is below in the individual responses. Second, thinking about limited recording bandwidth in neuroscience as a missing data problem is a core innovation of the paper which we did not emphasize previously. We clarify some crucial concepts in the individual responses to help the reviewers appreciate this aspect of the work. Third, many of the most widely used techniques in machine learning are based on simple ideas (e.g. dropout, L2 regularization) which benefit from generality and ease of implementation, so we do not believe this should be seen as a weakness. Fourth, we have now clarified that GPFA is a commonly used standard in the field and explained how it was more relevant than the suggested alternatives (e.g., LDS, Poisson-GPFA, or other dynamical interpolation methods). Finally, to address some of the concerns raised by the reviewers we now provide another point of comparison that illustrates both the generality of SB and the particular advantages of AutoLFADS+SBTT: we have added a new experiment applying SB to Neural Data Transformers (NDT). We show that SB also allows NDT to perform reasonably well with missing data, but that performance drops off much more quickly as a function of data sparsity. This suggests that SB is most effective with models that explicitly model the latent dynamics underlying the data. We thank the reviewers for helping us greatly improve the quality of our submission and look forward to receiving further feedback.

---

### Decision · Program_Chairs · 2021-09-28

**Decision:**

Accept (Poster)

**Comment:**

In the initial evaluations, all four reviewers recommended rejecting the paper, with the two primary reasons mentioned being that the methodological contributions of the paper are not sufficiently substantial to warranted publication at Neurips, and that comparisons (e.g. with interpolation methods) were missing. Even after the rebuttal phase, and internal discussions, reviewers decided to not change their opinion (and while not all might have updated their review, they did indicate so in internal discussions). Your AC does not necessarily agree with all of these criticisms, but does have to respect the consensus of the reviewers, and did not see sufficient grounds for overruling their unanimous opinion. I do hope that the feedback from the reviewers will allow you to further improve the manuscript.


**Consistency Experiment:**

NeurIPS has a long history of experimentation. In 2014, NeurIPS ran an experiment in which 10% of submissions were reviewed by two independent committees to quantify the randomness in the review process. This year, we repeated a variant of this experiment to see how the quality of the review process has changed over time.  This paper was part of the experiment and was therefore assigned to two committees (consisting of reviewers, an Area Chair, and a Senior Area Chair) that reached independent decisions.  If both committees made the same recommendation, this recommendation was followed. If a single committee recommended acceptance, the paper was accepted (with the exception of a few cases in which the other committee identified what we considered a fatal flaw, e.g., an error in a key result).

This copy’s committee reached the following decision: **Reject**

The other committee assigned to the paper recommended **Accept (Poster)**.  You can find the other set of reviews, along with any follow up discussion with the authors here:
https://openreview.net/forum?id=ZVrFO_Uru36